# Systematic Review of Peptide CAQK: Properties, Applications, and Outcomes

**DOI:** 10.3390/ijms252010990

**Published:** 2024-10-12

**Authors:** Jose A. Castillo, Michael Nhien Le, Amanda Ratcliff, Khadija Soufi, Kuanwei Huang, Sina Vatoofy, Arash Ghaffari-Rafi, Samuel Emerson, Elizabeth Reynolds, Christopher Pivetti, Kaitlin Clark, Allan Martin, Richard Price, Kee Kim, Aijun Wang, Rachel Russo

**Affiliations:** UC Davis Medical Center, Sacramento, CA 95817, USA; miqle@ucdavis.edu (M.N.L.); ajratcliff@ucdavis.edu (A.R.); khsoufi@ucdavis.edu (K.S.); kwehuang@ucdavis.edu (K.H.); ssvatoofy@ucdavis.edu (S.V.); aghaffarirafi@ucdavis.edu (A.G.-R.); sjemerson@ucdavis.edu (S.E.); elreynolds@ucdavis.edu (E.R.); cdpivetti@ucdavis.edu (C.P.); kcclark@ucdavis.edu (K.C.); armartin@ucdavis.edu (A.M.); riprice@ucdavis.edu (R.P.); kdkim@ucdavis.edu (K.K.); aawang@ucdavis.edu (A.W.);

**Keywords:** CAQK, central nervous system targeting, intravenous therapy

## Abstract

Many central nervous system (CNS) disorders lack approved treatment options. Previous research demonstrated that peptide CAQK can bind to chondroitin sulfate proteoglycans (CSPGs) in the extracellular matrix of the CNS. In vivo studies have investigated CAQK conjugated to nanoparticles containing therapeutic agents with varying methodologies/outcomes. This paper presents the first systematic review assessing its properties, applications, and outcomes secondary to its use. Following PRISMA guidelines, a comprehensive search was performed across multiple databases. Studies utilizing CAQK as a therapeutic agent/homing molecule in animal/human models were selected. Sixteen studies met the inclusion criteria. Mice and rats were the predominant animal models. All studies except one used CAQK to deliver a therapeutic agent. The reviewed studies mostly included models of brain and spinal cord injuries. Most studies had intravenous administration of CAQK. All studies demonstrated various benefits and that CAQK conjugation facilitated localization to target tissues. No studies directly evaluated the effects of CAQK alone. The data are limited by the heterogeneity in study methodologies and the lack of direct comparison between CAQK and conjugated agents. Overall, these findings present CAQK utilization to deliver a therapeutic agent as a promising targeting strategy in the management of disorders where CSPGs are upregulated.

## 1. Introduction

Central nervous system disorders are a major health and socio-economic problem, with a significant gap in pharmacological options that are currently approved for treatment [1] (p. 1). Traumatic brain injury (TBI) and spinal cord injury (SCI) are two prominent and prevalent examples that affect individuals of all ages. TBI is currently a leading cause of mortality in children, teens, and active adults from ages 1 to 44, with an annual incidence of 2.5 million in the US [2]. SCI has approximately 700,000 new cases diagnosed globally each year [3]. Both conditions, if not fatal, can lead to acute and potentially long-lasting neurological dysfunction [4]. Currently, treatment is limited to palliative care, and no specific therapies with long-term benefits are available [2,3]. Thus, there is a great need for advances in these fields.

The blood–brain barrier is considered a major impediment to the systemic treatment of these central nervous system diseases. Interestingly, in acute brain injury and SCI, along with several cerebrovascular diseases, including stroke, hypertension, and ischemia, this barrier is transiently disrupted, which allows extravascular access for macromolecules and neuroprotective drugs from systemic circulation [5]. However, the lack of specific binding of passively accumulating proteins in the injured area can result in low retention and subsequent washout over time. Due to limited localization and clearance over time, the therapeutic efficacy of a systemically administered drug is greatly limited.

Growing evidence suggests that an active targeting drug delivery system is one of the most promising therapeutic strategies for treating and managing TBI [6] and SCI [7,8]. Recent investigations reveal that the peptide CAQK, a molecule identified in 2016, selectively binds to extracellular matrix components present following central nervous system injury [9,10,11]. However, the literature remains limited, methodological approaches are heterogeneous, and CAQK’s properties and applications remain to be elucidated. To gain a better grasp of these questions, we sought to perform the first systematic review of the literature around CAQK and comprehensively cover its properties, current applications, and objective outcomes secondary to its use.

The CAQK peptide has attracted increasing attention from researchers due to its potential translational applications in therapeutic interventions. As the research surrounding CAQK advances, there is a need for a comprehensive evaluation of the existing literature to assess its translational potential and standardize clinical metrics/outcome measures. The purpose of this systemic review is to consolidate the current body of literature and propose considerations for future translational research to establish consistent and effective clinical evaluation frameworks for CAQK therapeutic use.

## 2. Materials and Methods

This systematic review was conducted in accordance with the Preferred Reporting Items for Systematic Reviews and Meta-Analysis (PRISMA) and the Cochrane Handbook of Systematic Reviews of Interventions. We included articles that met all relevant keywords and eligibility criteria aligned with the study’s objective. Our target topic focuses on reviewing the utilization of CAQK in translational models.

### 2.1. Search Strategy

A comprehensive search was performed to identify relevant studies and assess the transitional impact of CAQK on different injury models. Articles were obtained from MEDLINE, Embase, Scopus, and ScienceDirect databases, covering the period from inception to July 2024. We employed the search terms “CAQK” and “nanoparticles” as our primary search statement. Other than dates, no limits were utilized in the database search limitations. The initial search identified a total of 93 articles that were published from 2016 to 2024.

### 2.2. Eligibility Criteria

To refine our search, the inclusion criteria focused on papers that implemented CAQK as a therapeutic agent or homing molecule in animal or human studies. We applied the following eligibility criteria to each paper: (1) articles must be written in English; (2) articles must be relevant to the use of the CAQK peptide and its therapeutic potential; (3) articles must be randomized controlled trials on treatment outcomes in animals or human subjects; and (4) only medical or biomedical engineering publications are retained. Exclusion criteria were also established to ensure the quality and focus of the systemic review. These criteria include non-peer-reviewed publications, review articles, letters to the editor, meta-analyses, commentaries, conference abstracts, editorials, studies with insufficient clinical data, and duplicate publications across different databases. The screening and selection process for articles included in this systemic literature review are illustrated via a PRISMA flow diagram.

### 2.3. Data Extraction

Two reviewers independently reviewed all studies and independently extracted data from the studies using a predefined protocol. Any disagreement between the review authors was resolved through discussion. From the initial pool of 93 articles, a total of 16 articles were deemed eligible for full text review and inclusion in this systemic review. In accordance with recommendations from the *Cochrane Handbook for Systematic Reviews of Interventions* (chapter 5) [12], data were compiled into a shared Microsoft Excel spreadsheet. The collected data included the name of the paper, year of publication, the list of authors, and the publisher, as well as information pertinent to the systematic review including injury model, injury site, route of administration, dose, conjugated nanoparticle, therapeutic agent, cellular impact, functional assessment, and side effects. This structured approach ensured a comprehensive review of the available data and information on CAQK, focusing on its translational potential as a therapeutic strategy on injury models. The extracted data were analyzed and discussed with the aim of providing a deeper understanding and insight into the utilization, efficacy, safety, and future applications of CAQK.

### 2.4. Assessing Risk of Bias

The quality of studies selected in this systemic review was analyzed and determined by consensus among all authors using the Cochrane Collaboration tool [13]. Each domain was evaluated for risk bias and rated as “low”, “high”, or “unclear” with provided reasons to support their judgment. The summary of the quality assessment using this tool is summarized in Table 1.

## 3. Results

### 3.1. Informational Flow

The electronic database search yielded 93 citations. Thirty-six citations were removed due to duplication. After abstract screening and a full-text review of 57 unique citations, 16 studies met the eligibility criteria and were included in this study [3,5,10,11,14,15,16,17,18,19,20,21,22,23,24,25]. The study types included were all translational experiments. The removed citations were either a review article, duplicated article, not available to reviewers, or not a translational study. Figure 1 is a PRISMA flow diagram illustrating the above.

### 3.2. Animal Models

All studies reviewed were translational studies (Table 2). The years of publication were from 2016 to 2023. The countries of publication were either in the United States of America (USA) or China, with the majority coming from China. One article was from the country of Iran. Ten studies were conducted on mice and five studies on rats. No studies were performed on larger animals like a pig or monkey. Four mice studies used a TBI model, and nine studies used an SCI model that involved a weight drop in the thoracic spine of mice or rats. No SCI models involved the cervical spine. TBI models involved either a penetrating injury or a cortical impact. Two mice studies used a demyelination model that contained several sub-models, and one rat study looked at a knee cartilage defect model via a patellar disconnection.

The majority of studies involved the administration of an intravenous (IV) experimental therapy via, most commonly, the rat tail vein, apart from one SCI model using a hydrogel format directly applied to the injury site, and the knee cartilage defect model directly injecting the experimental therapy into the knee joint. One study administered their intravenous experimental therapy via the retro-orbital plexus. Figure 2 represents the different administration methods (intravenous, retro-orbital, intra-articular, and local) and injury models in pictorial form.

### 3.3. Human Correlations

Only one study performed an ex vivo experiment, using a brain sample from a patient who suffered a moderate TBI. Seven studies were performed in vitro, with five using SH-SY5Y cell lines, a human neuroblastoma cell, to demonstrate uptake and/or viability of the nanoparticle therapeutic agent conjugated to CAQK. The knee cartilage defect model also performed an in vitro study using human synovial fluid or cartilage to assess CAQK conjugated with the nanoparticle therapeutic agent for stability and/or cytotoxicity/uptake.

### 3.4. Conjugation/Target and Therapeutics

All but one article conjugated CAQK to a nanoparticle of some kind, including liposomal, silicon, and zein-based, polymeric micelles, and silver nanoparticles. Two articles specifically conjugated CAQK to an extracellular vesicle, and one conjugated it to a mitochondrion. Notably, one extracellular vesicle originated from a placental mesenchymal stem cell, while the other was from a neural stem cell/neural progenitor cell. Other nanoparticles used include a supramolecular assembly (HPAA-BM@CD-HPG-C), which was composed of a hyperbranched polymer-formed core/shell structure, and an activable protein nanoparticle (APNP). The study that conjugated CAQK directly to a mitochondrion identified the mitochondrion as the therapeutic agent.

Three studies assessed specifically what CAQK bound to. Mentioned proteins included versican, tenascin-R, and hyaluronan and proteoglycan link protein 4. The knee cartilage defect model mentioned studying a protein by the name of aggrecan. One article only tested the targeting ability of CAQK in a demyelination model. In the demyelination model, three sub-models were investigated with two of them being a cuprizone and experimental autoimmune encephalomyelitis model.

A wide variety of therapeutic agents were used ranging from apimaysin and arctigenin to methylprednisone and docetaxel, with only one duplicate therapeutic agent (metformin in two studies). Additionally, two studies did not utilize any therapeutic agent with their CAQK conjugated nanoparticle. Some studies utilized ribonucleic acids (RNAs) as their therapeutic agent, such as small interfering RNA (siRNA) or messenger RNA (mRNA). An example of a protein the mRNA encoded for was insulin-like growth factor 1 (IGF-1). Some studies directly utilized growth factors as their therapeutic agent, such as brain-derived neurotrophic factor in two and basic fibroblast growth factors in one. Figure 3 demonstrates the investigated and postulated binding targets of CAQK in pictorial form. Figure 4 demonstrates in pictorial form all the variety of nanoparticles, therapeutic agents, and their combinations conjugated to CAQK used in these studies, highlighting the diversity in therapeutic strategies involving CAQK and its conjugation to nanoparticles.

### 3.5. Efficacy

Most studies reported some form of efficacy with their nanoparticle therapeutic agent conjugated to CAQK (Table 2). No article compared their conjugated nanoparticle therapeutic agent directly with CAQK alone in terms of efficacy. Some articles compared their conjugated nanoparticle therapeutic agent to the therapeutic agent alone.

### 3.6. Dosing

The dosing of the experimental therapy was not found in five studies. Most reported only the dose used for the CAQK conjugated nanoparticle therapeutic agent. Dosing schedules ranged from once post injury to once weekly for 4 weeks post injury.

### 3.7. Cellular Level

On a cellular level, five articles demonstrated a smaller lesion size, and seven showed a reduction in inflammatory processes. This ranged to include generally more conversion of M1 to M2 macrophages, reduced pro-inflammatory cytokines like interleukins, and reduced reactive oxygen species. Additionally, seven articles showed neuroprotective features that included less neuron apoptosis and/or preserved axon density. In the knee cartilage model, it was found that there was less chondrocyte apoptosis.

### 3.8. Functional Level

Nine studies assessed locomotive abilities after administration of the experimental therapy post injury, all of which were performed in SCI models. Assessment methods included the Basso, Beattie, and Bresnahan or the Basso Mouse Scale open-field scoring system. Despite a variance in scoring methods, all studies showed improved hindlimb control. Additionally, five of these studies assessed motor evoked potentials to objectively record improvement. One of the TBI models also performed an elevate plus maze test that showed decreased anxiety, aggression, and dysphonia in the mouse. The duration of studies varied from less than 24 h to about 8 weeks. Of note, the studies with the longest recovery and study times involved the Basso, Beattie, and Bresnahan or the Basso Mouse Scale open-field scoring system, all of which were also performed in SCI models.

### 3.9. Side Effects

Side effect profiles were not assessed by all studies (Table 2). Those that did, performed hemolytic analyses and/or reviewed histology of harvested organs after administration of the experimental therapy. Hemolytic assessment varied but mainly included at most the following: aspartate aminotransferase (AST), alanine transaminase (ALT), blood urea nitrogen (BUN), serum creatinine (SCR), hemoglobin (HGB), platelets (PLT), and/or white blood cell count (WBC). Commonly harvested organs were heart, lung, liver, kidney, and/or spleen for assessment.

Six studies did not check for adverse reactions from their experimental therapy. Four showed no changes in the animal’s blood level values. Six studies found no histologic changes in the harvested organs after experimental therapy administration. An article that used metformin as their therapeutic agent reported liver/kidney histologic pathology at examination. One article that used apocynin as their therapeutic agent described a weight drop in their animals. They did go on to report that this returned to normal with time.

## 4. Discussion

### 4.1. Origin

Peptide CAQK was first identified by in vivo phage display screening in mice with acute brain injury and published in 2016 by Mann et al. The study utilized a T7 phage library that displayed on the phage surface nine amino acid cyclic peptides with the general composition of CX7C (C = cysteine; X = any amino acid) [26]. Following intravenous injection 6 hours post puncture brain injury, phage recovery was found to be 10 fold higher in the injured hemisphere compared to the uninjured contralateral side [10]. High-throughput sequencing analysis of the recovered phage pool revealed enrichment of phages with the tetra-peptide insert, CAQK (cysteine, alanine, glutamine, and lysine). CAQK was additionally not recovered from a control brain that came from a control mouse injected with the same phage library, indicating it has a specific role in the injured central nervous system.

This discovery marked the initial demonstration of CAQK’s capability to target central nervous system injury sites. In our systematic review, this foundational understanding was consistently reflected, as 15 out of 16 studies employed CAQK as a homing peptide conjugated to a nanoparticle therapeutic agent. Notably, just one study directly assessed CAQK to their CAQK conjugated nanoparticle therapeutic agent for targeting comparability only.

### 4.2. Protein Specificity

In the study by Mann et al., mass spectrometry proteomics analysis from extracts of injured brains with CAQK by affinity chromatography were performed to elucidate the favorable selectivity of CAQK to central nervous system injury. Immobilized peptides showed the proteins in eluates primarily belonging to the lectican family of chondroitin sulfate proteoglycans [27]. These mostly included versican, associated proteins tenascin-R (TNR), and the hyaluronan and proteoglycan link protein 4. It is known that in normal brains, lectican proteoglycans form extracellular matrix complexes, and the expression of some of these lectican proteoglycans is upregulated at sites of central nervous system injury [28,29]. Mann et al. confirmed this finding through immunostaining and demonstrated that the expression of extracellular matrix-associated chondroitin sulfate proteoglycans (versican, TNR, and hyaluronan and proteoglycan link protein 4) were all upregulated following injury when compared to the uninjured hemisphere of the brain.

Our review identified two studies specifically evaluating CAQK binding targets. One study used a demyelination model, while the other focused on a knee cartilage defect model. In the demyelination model, three sub-models were investigated. Fluorescently labeled CAQK was confirmed to have targeted fibrous extracellular material at lesion sites when injected into the circulation in both the cuprizone and experimental autoimmune encephalomyelitis models, TNR was observed at lower abundance in lesion sites compared to surrounding healthy tissue, and no significant peptide overlap with TNR was noted. Conversely, tenascin-C (TNC) deposits were variably present within demyelinating lesions in the two models. In the experimental autoimmune encephalomyelitis model, but not in the cuprizone model, CAQK localized to fibrous TNC deposits suggesting that its target sites include epitopes in the local extracellular matrix; however, it is important to note that most of the peptide’s target area in lesions lacked TNC expression. The data suggest the open nature of the peptide to interact with molecules at sites of demyelination and that this might vary during the dynamic progression of de- and remyelination. These articles indicate that the true targeted protein may potentially occur through TNR/TNC/hyaluronan and proteoglycan link protein 4 but can also include edited forms or a combination with other epitopes in the extracellular matrix, which may vary with different types and stages of central nervous system injury. Further research is needed to elucidate the precise nature of the peptide’s binding partners and their relevance to different stages of demyelination and remyelination.

This missing understanding holds relevance regarding CAQK’s ability to impact effects, as it was generally seen that if the extracellular matrix was not altered, then CAQK’s potential impact did not occur. To demonstrate the versatility of CAQK, Mann et al. tested the delivery of oligonucleotides loaded into porous silicon nanoparticles as a carrier [30]. Their proof of concept approach was to silence the local expression of green fluorescent protein systemically expressed in transgenic mice from the CAG promoter [31]. They simulated therapeutic oligonucleotides by using short interfering RNA against green fluorescent protein loaded in CAQK conjugated porous silicon nanoparticles. Porous silicon nanoparticles were intravenously injected into the green fluorescent protein mice with penetrating brain injury and visualized by time-gated luminescence imaging [32,33], allowing quantification of their accumulation in the excised brains. The imaging showed that CAQK porous silicon nanoparticles accumulated in the injuries at markedly higher (35-fold) levels than porous silicon nanoparticles coated with a control peptide. They found through confocal microscopy on transverse cortical sections from mice injected with CAQK porous silicon nanoparticles that short interfering green fluorescent protein exhibited a large void of green fluorescent protein expression at the injury site, whereas brains from mice treated with control nanoparticles did not differ from untreated brains. They conclude that this gene silencing was specific for brain injury, as green fluorescent protein expression remained unaltered in normal brains. The few other articles that did the same as this one demonstrated the extracellular matrix’s relevance to CAQK’s ability to bind and in turn cause effect.

### 4.3. CAQK Applications

Mann et al. assessed CAQK in additional injury models, including skin and liver, where only a marginal increase in binding was observed compared to the TBI model. Ultimately, the accumulation was noted to be far less than that of the TBI site [10]. As mentioned before, we found that only three articles used CAQK in a TBI model, with the remaining (10, 2, and 1) assessing its role in SCI, demyelination, or knee cartilage defect models, respectively. The efficacy observed in SCI models is consistent with the role of chondroitin sulfate proteoglycans that have been shown to be crucial to the SCI recovery process by inhibiting neural regeneration or enclosing the damaged area [9,34] and can substantially manifest at the site of injury within 24 h after and even persist for several months [35,36].

Despite the original study’s initial assertion that CAQK targets specific chondroitin sulfate proteoglycans, findings from the demyelination models suggest that other targets upregulated in the extracellular matrix and arising with TNR/TNC can possibly be at play. They also demonstrated the possibility of this phenomenon being impacted by the disease process in question. If multiple combinations of TNR/TNC attract CAQK, there may be a role for this molecule in other disease or injury models. This was demonstrated by the study that showed a potential application of CAQK in a knee cartilage defect model and the role of aggrecan as a target, a known member of the lectican family that is abundant in cartilage and shares the same structural domains at both N and C-termini with versican. They showed that fluorescence-labeled lipid nanoparticles confirmed CAQK’s affinity for aggrecan, and following the removal of unbound nanoparticles, a higher level of fluorescent intensity in wells with aggrecan was detected compared to nontargeted lipid nanoparticles and wells without aggrecan, which resulted in decreased retention of CAQK-bound lipid nanoparticles.

In reviewing the demyelination model and the cartilage model, it becomes clear that the fundamental understanding of what specifically CAQK binds to needs elucidating further. If TNR/TNC or some combination with a familiar epitope are not unique to the central nervous system, future studies should be mindful of CAQK’s use in a polytrauma model, for example, as this could complicate result interpretation.

### 4.4. Benefits/Side Effects

Our review highlighted that CAQK effectively aided in targeting the desired injury site and conferred therapeutic benefits across nearly all the studies. The benefits ranged from macroscopically measured outcomes, for example, improved motor evoked potentials or some form of motor scoring system, to cellularly measured outcomes, including reduced pro-inflammatory markers to neuroprotection and axonal preservation. Decreased central nervous system lesion size was also noted in five of the studies evaluated. Unfortunately, all studies assessed for these utilized a nanoparticle therapeutic agent conjugated to CAQK, and none directly compared it to CAQK administration only, in regard to benefits/side effects, making any single benefit/side effect difficult to decipher.

Most benefits or side effects appeared to be unique to the attached respective therapeutic. This is deduced given that no two studies yielded similar results at study conclusion. Regarding side effects, several studies investigated the accumulation/excretion of CAQK conjugated to the nanoparticle therapeutic agent in various other organs. Mann et al. originally showed that no comparable amount of CAQK accumulation was observed in the lung, liver, or skin, after 30 min of circulation, except for the kidney, which is recognized as a primary site for peptide clearance. This observation was consistently corroborated by subsequent studies using real-time imaging or post-experiment organ harvesting and examination. However, variations in organ accumulation/excretion were observed depending on the nanoparticle therapeutic agent conjugated to CAQK. For instance, two studies employing metformin as a therapeutic agent reported higher liver accumulation compared to the kidney, aligning with metformin’s known hepatic action [37]. In one of those studies, they also go on to describe less liver pathology on histological examination. These findings are interesting because it shows the ability of using nanoparticles to minimize the toxic effects of a therapeutic agent. This fact is consistent with the literature. Lee et al. demonstrated that a high dose of free methylprednisolone (100 mg/kg) and a low dose of methylprednisolone loaded in nanoparticles (10 mg/kg) have the same effect in reducing astrocyte activation in the spinal cord-injured mice. Therefore, their results showed that the nanoparticle was able to reduce the amount of required drug for diminishing the inflammation [38].

It remains unclear if CAQK collection is affected by nanoparticle therapeutic agent attachment, as only these studies directly assessed and compared CAQK to their CAQK conjugated nanoparticle therapeutic agent group. This raises questions regarding whether the nanoparticle therapeutic agent itself can affect the distribution and excretion of CAQK, and if this is dependent on the concentration or mass ratio of CAQK and the nanoparticle therapeutic agent. This variability could potentially pose a unique predicament, should, for example, CAQK bound to a nanoparticle therapeutic agent become excreted in an organ that is at baseline pathology, resulting in undesired outcomes that could negate a potential therapeutic benefit. Future research studies should account for these variables when comparing treatment modalities involving CAQK to help interpretations of results related to CAQK distribution/excretion and minimize confounding factors.

The duration of studies also varied widely, from hours to weeks before animal euthanasia resulting in numerous limitations such as potential placebo effects, which have been documented in animal studies as well [39]. Elaboration on the reasoning for each study’s respective duration was not given in any study. This variability highlights the need for careful consideration in future experiment design to additionally assess the long-term effects and efficacy of CAQK-influenced therapies.

### 4.5. Dosing

CAQK dosing and frequency of administration varied greatly between studies due to the variation in conjugated nanoparticle therapeutic agents. CAQK effectively facilitated the targeting of therapeutic agents to a site of injury on the spinal cord or brain in concentrations ranging from 1 nmol/µL to 2.23 nmol/µL. This range was reverse calculated as most articles only listed doses of CAQK bound to a nanoparticle therapeutic agent. This was only performed in rodents, either mice or rats, with none performing their experiment on larger animals. Dosing schedules also varied greatly from injection once post injury to injection once weekly for 4 weeks post injury [3,15]. As seen with study duration, elaboration on the reasoning for each studies respective dosing schedule was not given in any study. Future studies should consider incorporating dosing group comparisons to aid in identifying thresholds at which a benefit becomes noticeable and identify optimal dosing thresholds that maximize therapeutic benefit while minimizing dosage. Larger animal experiments could also help to understand the potential dosing needed in humans as it is likely that far more would be used in the latter.

### 4.6. Human Targeting

Seven studies investigated CAQK’s potential to target and impact human tissue. One study assessed human cartilage, one assessed the human brain, and seven assessed other human tissues. The other human tissue included astrocytoma cells, neuroblastoma cells, and synovial fluid.

In this study, in vitro binding of the CAQK phage to the extracellular matrix produced by U251 human astrocytoma cells was tested. These cells express high levels of versican and other members of the brain extracellular matrix [40], which suggests that these cells are activated in culture. They reported that there was significantly higher binding to the U251 extracellular matrix being shown when compared to a control phage. They noticed that binding to this extracellular matrix was specific, and it was inhibited with excess free CAQK peptide. Moreover, after enzymatic treatment of the extracellular matrix with chondroitinase ABC or hyaluronidase, enzymes that break down chondroitin sulfate proteoglycans, this resulted in loss of versican staining and correspondingly reduced CAQK binding. This suggests that the epitope for CAQK resides in the complex formed by the chondroitin sulfate proteoglycans, hyaluronic acid and associated proteins, as seen in the mouse brain.

This was also demonstrated ex vivo by CAQK conjugated to a nanoparticle binding to a human cortical brain section that had suffered a moderate TBI. The CAQK conjugated to nanoparticles showed intense binding to the injured brain sections from the cortex and the corpus callosum areas, whereas binding to normal brain sections was minimal. Similar to the mouse brains, we observed significant elevation in the expression of versican and hyaluronan and proteoglycan link protein 4 in injured brains compared with normal brains by immunohistochemistry.

These findings are not surprising, as peptides are generally not species specific in their binding properties [41]. It is known that ex vivo methods can offer more physiologically relevant insights into the whole organism because they maintain some native interactions with tissues or organs [15].

These preliminary findings indicate that CAQK has a potential therapeutic role in humans, but further research is necessary for more robust conclusions. Future studies should involve both ex vivo or in vitro models using the respective human tissue counterparts to provide more definitive insights into CAQK’s efficacy.

### 4.7. Assessment and Risk of Bias

Any definitive statement is hard to make with a systematic review as it is an interpretation of numerous articles by a reviewer. What one captures for review is limited to what is reported. All these articles published a positive result leading to a publication bias that affects the collective literature on animal experiments using CAQK. They all report the ability of CAQK to home their experimental agent to an injury site. This gives the notion that CAQK is only useful as a homing peptide. As mentioned before, a confounding bias cannot be excluded as CAQK was never tested individually and compared against the groups that conjugated CAQK to their nanoparticle containing a therapeutic agent for outcomes. When one experiments, for example, with an experimental agent in saline, a group is normally devised that receives only saline, to rule out any direct impact of saline administration. When this is not carried out, it makes it unclear if any baseline direct impacts, in this case from CAQK, are occurring and unknowingly adjusting outcomes.

### 4.8. Future Perspectives

Moving forward, studies should practice a standard when experimenting with CAQK. They should consider a group that receives CAQK only to compare with the group conjugating it to a nanoparticle containing a therapeutic agent to rule out any direct confounding impacts from CAQK alone. As was carried out in the original paper, affinity chromatography, or a similar test, should be performed to elucidate the favorable selectivity of CAQK in the injury model of said study to help define what exactly CAQK is binding to. Dosing group comparisons should be implemented as well to try and elucidate an optimal dosing threshold at which effects are shown, and lastly, ex vivo or in vitro analyses of respective human tissue counterparts for specific animal injury models should be performed to confirm translatability to human subjects.

## 5. Conclusions

Sixteen studies exist demonstrating CAQK’s role in the targeting the central nervous system and knee cartilage injury sites. Overall, almost all studies showed benefits in CAQK nanoparticle therapeutic agent administration ranging from functional changes like improved locomotion to cellularly via neuroprotection and anti-inflammation. It is unclear if this is secondary to or confounded by CAQK given that no direct comparisons were made since most studies used it as a homing protein. The fundamental understanding of its targeting mechanism of action remains to be explained given the efficacy in a cartilage model and the theoretical possibility of CAQK being attracted to other organs with chondroitin sulfate proteoglycan upregulation during injury.

This literature review shows the early stages of scientific discovery unfolding, and future studies should aim to practice a standard when experimenting with CAQK to aid in deciphering its many unknowns.

## Figures and Tables

**Figure 1 ijms-25-10990-f001:**
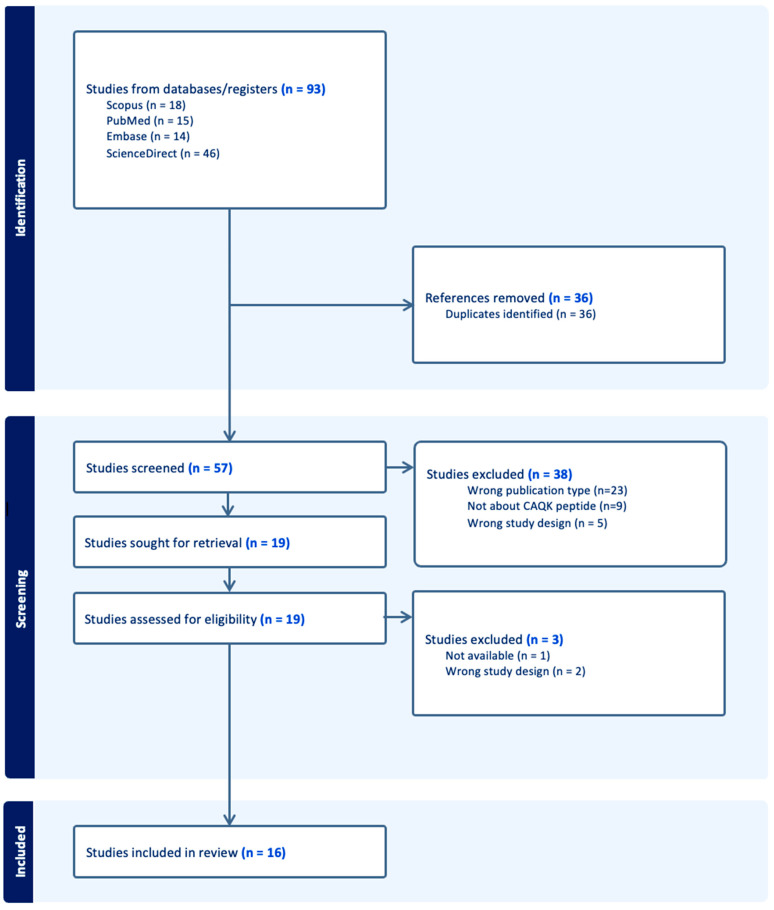
PRISMA flow diagram.

**Figure 2 ijms-25-10990-f002:**
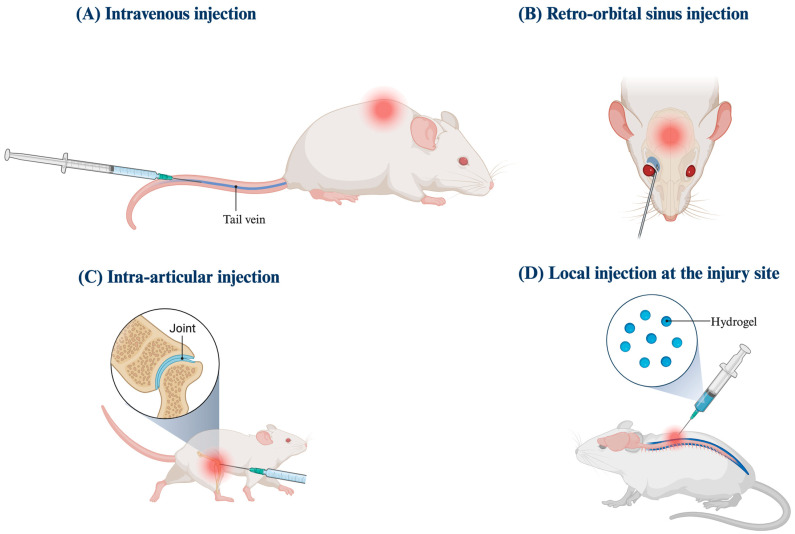
Injury models and experimental therapy routes of administration. (**A**) Intravenous injection administered via the tail vein in an SCI model. (**B**) Retro-orbital sinus injection in a TBI model. (**C**) Intra-articular injection into the knee joint space in a patellar disconnection model. (**D**) One study utilized local injection at the injury site via a hydrogel application directly to the site of SCI. Created with BioRender.com (accessed on 9 October 2024).

**Figure 3 ijms-25-10990-f003:**
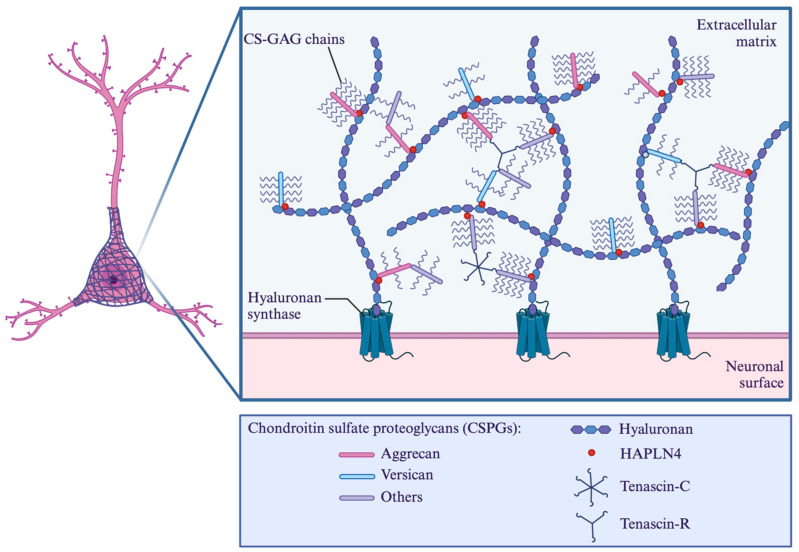
Investigated and postulated CAQK binding sites. A schematic illustration of the extracellular matrix components and their relationship with neuronal surfaces, focusing on potential CAQK binding sites (HAPLN4—hyaluronan and proteoglycan link protein 4; CS-GAG—chondroitin sulfate-glycosaminoglycan). Created with BioRender.com (accessed on 9 October 2024).

**Figure 4 ijms-25-10990-f004:**
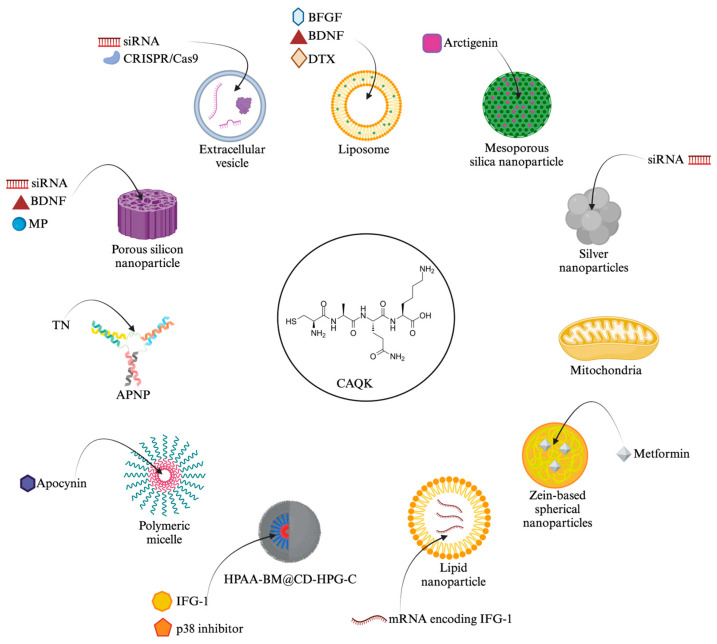
CAQK conjugated nanoparticles and investigated therapeutic agents. An overview of various CAQK-conjugated nanoparticle systems, designed for delivering different therapeutic agents that have been used in experimental studies (BFGF—basic fibroblast growth factor; BDNF—brain-derived neurotrophic factor; DTX—docetaxel; si/mRNA—small interfering/messenger ribonucleic acid; IFG-1—insulin-like growth factor 1; TN—Tat-NR2BAc; APNP—activatable protein nanoparticle; MP—methylprednisolone). Created with BioRender.com (accessed on 9 October 2024).

**Table 1 ijms-25-10990-t001:** Cochrane Collaboration’s tool for assessing risk of bias (“+”, low risk of bias).

Bias Domain	Source of Bias	AP Mann [10]	Q Wang [14]	G Sun [5]	P Wu [15]	J Wang [11]	C Abi-Ghanem [16]	T Li [17]	T Li [18]	Y Rong [19]	LE Waggoner [20]	B Wang [21]	F Wu [3]	J Xu [22]	P Xu [23]	X Yu [24]	L Zare [25]
Selection Bias	Random Sequence Generation	+	+	+	+	+	+	+	+	+	+	+	+	+	+	+	+
Allocation Concealment	+	+	+	+	+	+	+	+	+	+	+	+	+	+	+	+	+
Performance Bias	Blinding of participants and personnel	+	+	+	+	+	+	+	+	+	+	+	+	+	+	+	+
Detection Bias	Blinding of outcome assessment	+	+	+	+	+	+	+	+	+	+	+	+	+	+	+	+
Attrition Bias	Incomplete outcome data	+	+	+	+	+	+	+	+	+	+	+	+	+	+	+	+
Reporting Bias	Selective reporting	+	+	+	+	+	+	+	+	+	+	+	+	+	+	+	+
Other Bias	Anything else, ideally prespecified	+	+	+	+	+	+	+	+	+	+	+	+	+	+	+	+

**Table 2 ijms-25-10990-t002:** All translational studies utilizing CAQK (PI—post injury; si/mRNA—small interfering/messenger ribonucleic acid; CBC—complete blood count; BMP—basic metabolic panel; BBB—Basso, Beattie, and Bresnahan scoring system; MEP—motor evoked potential; BMS—Basso Mouse Scale open-field scoring system).

Reference	Injury Model	Mechanism of Injury	Route of Administration	Treatment Frequency	Nanoparticle	Therapeutic Molecule	CAQK Concentration	Cellular Impact	Functional Impact	Adverse Effect
1. AP Mann et al., 2016 USA [10]	TBI	Penetrating injuryBlunt cortical impact	Tail vein	6 h and 24 h PI	Porous silicon nanoparticles (PSiNPs)Silver nanoparticles	siRNA	300 μg of CAQK-PSiNPs/siRNA	Gene downregulation	N/A	N/A
2. P Wu et al., 2019 China [15]	TBI	Controlled cortical impact	Tail vein	~PI, Once	Activatable protein nanoparticle (APNP)	Tat-NR2B9c (TN)	N/A	Neuroprotective	Improved EPM	Not tested
3. LE Waggoner et al., 2022 USA [20]	TBI	Controlled cortical impact	Tail vein	2 h PI, Once	Porous silicon nanoparticle (PSiNPs)	Brain-derived neurotrophic factor	10 μg of CAQK	Neuroprotective	N/A	N/A
4. L Zare et al., 2023 Iran [25]	TBI	Demyelination	Tail vein	6d PI, Once	Porous silicon nanoparticle (PSiNPs)	Methylprednisolone (MP)	4.84 μg of CAQK-PSiNPs/MP	Anti-inflammatoryNeuroprotective	N/A	N/A
5. C Abi-Ghanem et al., 2022 USA [16]	TBISCI	Demyelination	Retro-orbital plexusTail vein	24 h or 5d PI, Once	None	None	1 nmol/μL of FAM-CAQK	N/A	N/A	N/A
6. Q Wang et al., 2018 China & USA [14]	SCI	Weight drop	Local administration using hydrogel	~PI, Once	Liposome (LIP)	Docetaxel (DTX)Brain-derived neurotrophic factor (GFs)	10 μg of CAQK-LIP/GFs & DTX	Neuroprotective	Improved BBB and footprint test	N/A
7. G Sun et al., 2019 China [5]	SCI	Weight drop	Tail vein	1d PI, Once	Mesoporous silica nanoparticles (MSN)	Arctigenin (ARC-G)	0.5 mg of CAQK-MSN/ARC-G(on uninjured model)	Anti-inflammatoryNeuroprotective	Improved MEP	No adverse effects in heart, liver, spleen, lung, or kidneyCBC/BMP normal
8. J Wang et al., 2020 China [11]	SCI	Weight drop	Intravenous	2 h then every 2d PI	Polymeric micelle	Apocynin (APO)	13.388 nmol/μL of CAQK	Anti-inflammatoryNeuroprotective	Improved BMS and footprint test	No adverse effects in heart, liver, spleen, lung, or kidney.
9. T Li et al., 2022 China [17]	SCI	Weight drop	Intravenous	6 h PI, Once	Zein-based spherical nanoparticles (NPs)	Metformin (MET)	300 μg of CAQK	N/A	N/A	N/A
10. T Li et al., 2022 China [18]	SCI	Weight drop	Tail vein	6 h PI then daily for 7d	Zein-based spherical nanoparticles (NPs)	Metformin (MET)	N/A	Anti-inflammatoryNeuroprotective	Improved MEP	Liver and kidney damageMyelosupression that resolved
11. Y Rong et al., 2022 China [19]	SCI	Weight drop	Tail vein	daily for 5d PI	Extracellular vesicles (EVs)	siRNA	1 μg/μL CAQK-EVs	Anti-inflammatoryNeuroprotective	Improved MEP	No adverse effect
12. B Wang et al., 2022 China [21]	SCI	Weight drop	Tail vein	~PI, Once	PMSC-EVs (EXO)	CRISPR/Cas9 (@P)	50 μg of CAQK-EXO/@P	Anti-inflammatoryNeuroprotective	Improved BMS and CatWalk	No adverse effects in heart, liver, spleen, lung, or kidneyCBC/BMP normal
13. F Wu et al., 2023 China [3]	SCI	Weight drop	Tail vein	Weekly for 4 weeks	Dual-targeting liposome with R2KC peptide	Basic fibroblast growth factor	N/A	Anti-inflammatoryNeuroprotective	Improved BBB and footprint test	N/A
14. J Xu et al., 2023 China [22]	SCI	Weight drop	Tail vein	3d PI	Mitochondria (Mito)	Mitochondria (Mito)	1 mg of Tpp-CAQK-Mito	Anti-inflammatoryNeuroprotective	Improved MEP, BMS, LSS, grid walking test	No adverse effect in heart/liver/spleen/lung/kidney
15. P Xu et al., 2023 China [23]	SCI	Weight drop	Intravenous	2 h then daily for 7d PI	HPAA-BM@CD-HPG-C	p38 inhibitorIGF-1	N/A	Neuroprotective	Improved MEP and BMS	No adverse effects in heart, liver, spleen, lung, or kidneyCBC/BMP normal
16. X Yu et al., 2023 China [24]	Knee joint	Patellar disconnection	Local	~PI, Once	Cartilage targeting ionizable lipid nanoparticle	IGF-1 mRNA	N/A	Anti-apoptosis	N/A	No adverse effects in heart, liver, spleen, lung, or kidneyCBC/BMP normal

## Data Availability

No new data were created or analyzed in this study. Data sharing is not applicable to this article.

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
