# Peer review of "Systematic Review of Peptide CAQK: Properties, Applications, and Outcomes"

_ijms, 2024, doi:10.3390/ijms252010990_

Round 1
Reviewer 1 Report
Comments and Suggestions for Authors
The general overview of articles published on CAQK presented by the authors is interesting. However, several questions arise from the manuscript's analysis. It isn't easy to understand why, out of 93 studies, the analysis is based on 16 reports. In the screening in Figure 2 and the text, there is no clear information as to why 38 studies were excluded. Table 1 is informative but challenging to understand . The articles should probably be grouped by the model or the therapeutic molecule. Why the Table comes before the Figure 2? The analysis of human correlation is weak, it should be more informative. Figure 3 is informative, but there is not much discussion on the relevance of the extracellular matrix in the process and the decrease in pharmacological effectiveness. The discussion of Figure 4 is also concise based on the fact that the carriers must not interact with the compound of interest; some might. Finally, there should be a discussion on future perspectives based on the data.
Author Response
Reviewer 1:
Comment 1: The general overview of articles published on CAQK presented by the authors is interesting. However, several questions arise from the manuscript's analysis. It isn't easy to understand why, out of 93 studies, the analysis is based on 16 reports. In the screening in Figure 2 and the text, there is no clear information as to why 38 studies were excluded.
Author’s response: Thank you very much for your attention to our manuscript, we greatly appreciate your feedback and suggestion. In response, we have modified Figure 1 to be the PRISMA flow diagram. It now also has specified information on why we excluded 38 studies. Additionally, this is also mentioned in the new text for clarity. (Lines 126-128)
Change to Text: The electronic database search yielded 93 citations. Thirty-six citations were removed due to duplication. After abstract screening and full-text review of 57 unique citations, 16 studies met the eligibility criteria and were included in this study. The study types included were all translational experiments. The removed citations were either a review article, duplicated article, not available to reviewers, or not a translational study. Figure 1 is a PRISMA flow diagram illustrating the before mentioned.
Comment 2: Table 1 is informative but challenging to understand. The articles should probably be grouped by the model or the therapeutic molecule. Why the Table comes before the Figure 2?
Author’s response: Thank you for your response. We reorganized the table and incorporated some changes for clarity. We have grouped the articles by the injury model to make it easier to understand.
Comment 3: The analysis of human correlation is weak, it should be more informative.
Author’s response: Thank you for your feedback. We agree and have elaborated on CAQKs ability to interact with human tissue in more detail. We now describe the paper speaking on U251 human astrocytoma cell testing in more detail. We included more about their methods and how they specifically showed that when chondroitin sulfate proteoglycans (CSPGs) are degraded there is less binding of CAQK, as was seen in the mouse model. We also expanded on the information around how CAQK conjugated to a nanoparticle, which could be carrying a therapeutic agent, is noted to be less bound to non-traumatized tissue. On further investigation by the authors of the discovery paper, this was due to less CSPG presence, a general theme in the reviewed articles that looked at this notion as well. We also added a reference, that proves similar peptide biding results can be noted between species, supporting the similar results appreciated in mouse and human brain interacting with CAQK as not surprising. Lastly, with this text addition it now also strengthens the second to last sentence in the discussion, that it is within reason to infer that CAQK has a potential to impact human tissue. (Lines 432-434, 436-442, 444-451)
Change to Text: In the discovery paper, in vitro binding of CAQK phage to extracellular matrix produced by U251 human astrocytoma cells was tested. These cells express high levels of versican and other members of the brain extracellular matrix27, which suggests that these cells are activated in culture. They reported that there was significantly higher binding to the U251 extracellular matrix being appreciated when compared to a control phage. They noticed binding to this extracellular matrix was specific as well as it was inhibited with excess free CAQK peptide. Moreover, after enzymatic treatment of the extracellular matrix with chondroitinase ABC or hyaluronidase, enzymes that break down chondroitin sulfate proteoglycans, this resulted in loss of versican staining and correspondingly reduced CAQK binding. This suggests that the epitope for CAQK resides in the complex formed by the chondroitin sulfate proteoglycans, hyaluronic acid and associated proteins, as seen in the mouse brain. This was also demonstrated ex vivo by CAQK conjugated to a nanoparticle binding to a human cortical brain sections that had suffered a moderate TBI. The CAQK conjugated to nanoparticles showed intense binding to the injured brain sections from the cortex and the corpus callosum areas, whereas binding to normal brain sections was minimal. Similar to the mouse brains, we observed significant elevation in expression of versican and hapln4 in injured brain than in normal brain by immunohistochemistry. These findings are not surprising, as peptides are generally not species-specific in their binding properties28. It is known that ex vivo methods can offer more physiologically relevant insights into the whole organism because they maintain some native interactions with tissues or organs26.
Comment 4: Figure 3 is informative, but there is not much discussion on the relevance of the extracellular matrix in the process and the decrease in pharmacological effectiveness.
Author’s response: Thank you for your feedback. We agree and have added a paragraph in the section in the discussion titled protein specificity. With this text change we speak to how CAQKs ability to affect is dependent on the extracellular matrix (ECM) changes post injury. (Lines 307-326)
Change to Text: This missing understanding holds relevance to CAQKs ability to impact effects, as it was generally seen that if extracellular matrix was not altered then CAQKs potential impact did not occur. To demonstrate the versatility of CAQK, Mann et. Al. tested delivery of oligonucleotides loaded into porous silicon nanoparticles as a carrier16. Their proof of concept approach was to silence local expression of green fluorescent protein systemically expressed in transgenic mice from the CAG promoter17. They simulated therapeutic oligonucleotides by using short interfering RNA against green fluorescent protein loaded in CAQK conjugated porous silicon nanoparticles. Porous silicon nanoparticles were intravenously injected into the green fluorescent protein mice with penetrating brain injury and visualized by time-gated luminescence imaging18,19 allowing quantification of their accumulation in the excised brains. The imaging showed that CAQK porous silicon nanoparticles accumulated in the injuries at markedly higher (35 fold) levels than porous silicon nanoparticles coated with a control peptide. They found that confocal microscopy on transverse cortical sections from mice injected with CAQK porous silicon nanoparticles short interfering green fluorescent protein exhibited a large void of green fluorescent protein expression at the injury site, whereas brains from mice treated with control nanoparticles did not differ from untreated brains. They conclude that this gene silencing was specific for brain injury, as green fluorescent protein expression remained unaltered in normal brains. The few other articles that did as this one, are demonstrating extracellular matrix’s relevance to CAQKs ability to bind and in turn cause effect.
Comment 5: The discussion of Figure 4 is also concise based on the fact that the carriers must not interact with the compound of interest; some might.
Author’s response: Thank you for your remarks. The discussion now speaks more to the possibility of different benefits/side effects taking place when a nanoparticle is used to transport a certain therapeutic agent. We now describe 2 articles that support this. The one from our reviewed articles appreciated this interaction when they compared CAQK conjugated nanoparticle containing metformin to just metformin alone. They found that it has less toxic impact on the liver when contained within the nanoparticle. The other article cited from one of our reviewed articles found that the nanoparticle could possibly impact the dose needed for the administered therapeutic agent to have effect. In this citation they placed high and low dose methylprednisolone into separate nanoparticles. They found the same effect in terms of astrocyte activation reduction with the varying doses in their multiple sclerosis model suggesting the nanoparticle reduced the required therapeutic agent amount to cause the effect. (Lines 396-400)
Change to Text: The study utilizing metformin, in example, compared it to the nanoparticle metformin conjugated to CAQK and appreciated less liver pathology. Lee et al. showed that a high dose of free methylprednisolone (100 mg/kg) and a low dose of methylprednisolone loaded in nanoparticles (10 mg/kg) have the same effect in reducing astrocyte activation in the spinal cord-injured mice. Therefore, their results showed that the nanoparticle was able to reduce the amount of required drug for diminishing the inflammation24. Future research studies should account for these variables when comparing treatment modalities involving CAQK to minimize confounding factors.
Comment 6: Finally, there should be a discussion on future perspectives based on the data.
Author’s response: Thanks for your remark. We have added a section titled future perspectives in the discussion just prior to the conclusion. (Lines 474-484)
Change to Text:
4.8. Future Perspectives
Moving forward, studies should practice a standard when experimenting with CAQK. They should consider a group that receives CAQK only to compare with the group conjugating it to a nanoparticle containing a therapeutic agent to rule out any direct confounding impacts from CAQK alone. As done in the original paper, affinity chromatography, or a similar test, should be performed to elucidate the favorable selectivity of CAQK in the injury model of said study to help define what exactly CAQK is binding to. Dosing group comparisons should be implemented as well to try and elucidate at an optimal dosing thresholds at which effects are appreciated, and lastly, ex vivo or in vitro analyses of respective human tissue counterpart for specific animal injury model should be done to confirm translatability to human subjects.
Reviewer 2 Report
Comments and Suggestions for Authors
To whom it may concern,
The current manuscript needs to fulfill the requirement for a systematic review according to the 2020 PRISMA checklist. Please consider/correct the following aspects:
1. Rewrite the abstract according to the Prisma checklist for abstracts.
2. Provide clearly in the introduction the aims of the systematic review.
3. The second chapter should be the "materials and methodology" part. I suggest that in this part, you create subchapters that address the following issues: eligibility criteria, data collection process, risk of bias, and precise measures.
4. The results section should be improved. Again, you have no data on the assessment and risk of bias in this part.
5. The reference list is short. Try to improve with more significant studies, particularly in the discussion section.
6. If you do not think this paper can address totally the PRISMA requirement, then you can change the type of article from systematic review to scoping review. Still, my suggestions are relevant and would improve the quality of the present paper.
Comments on the Quality of English LanguageOnly minor changes are needed, quality of language is good.
Author Response
Reviewer 2:
Comment 1: The current manuscript needs to fulfill the requirement for a systematic review according to the 2020 PRISMA checklist. Please consider/correct the following aspects: Rewrite the abstract according to the Prisma checklist for abstracts.
Author’s response: Thank you for your response. We have incorporated this suggestion in our newly written abstract. (Lines 13-27)
Change to Text: Many central nervous system (CNS) disorders lack approved treatment options. Previous research demonstrated peptide CAQK can bind to chondroitin sulfate proteoglycans (CSPGs) in the extracellular matrix of the CNS. In vivo studies have investigated CAQK conjugated to nanoparticles containing therapeutic agents with varying methodologies/outcomes. This paper presents the first systematic review assessing its properties, applications, and outcomes secondary to its use. Following PRISMA guidelines, a comprehensive search was performed across multiple databases. Studies utilizing CAQK as a therapeutic agent/homing molecule in animal/human models were selected. Sixteen studies met the inclusion criteria. Mice and rats were the predominant animal models. All studies except 1 used CAQK to deliver a therapeutic agent. The reviewed studies mostly included models of brain and spinal cord injuries. Most studies had intravenous administration of CAQK. All studies demonstrated various benefits and that CAQK conjugation facilitated localization to target tissues. No studies directly evaluated the effects of CAQK alone. The data is limited by the heterogeneity in study methodologies and the lack of direct comparison between CAQK and conjugated agents. Overall, these findings present CAQK utilization to deliver a therapeutic agent as a promising targeting strategy in the management of disorders where CSPGs are upregulated.
Comment 2: Provide clearly in the introduction the aims of the systematic review.
Author’s response: We have more clearly defined our aims in the introduction. (Lines 59-65)
Change to Text: CAQK peptide has attracted increasing attention from researchers due to its potential translational applications in therapeutic interventions. As the research surrounding CAQK advances, there is a need for a comprehensive evaluation of existing literature to assess its translational potential and standardize clinical metrics/outcome measures. The purpose of this systemic review is to consolidate the current body of literature and propose considerations for future translational research to establish consistent and effective clinical evaluation frameworks for CAQK therapeutic use.
Comment 3: I suggest that in this part, you create subchapters that address the following issues: eligibility criteria, data collection process, risk of bias, and precise measures.
Author’s response: Thank you for the feedback. The methods are now dramatically improved thanks to your suggestions, and we have incorporated them into the revised manuscript. An assessment and risk bias section has also been added to the discussion. (Lines 67-118)
Change to Text:
- Materials and Methods
This systematic review was conducted in accordance with the Preferred Reporting Items for Systematic Reviews and Meta-Analysis (PRISMA) and the Cochrane Handbook of Systematic Reviews of Interventions. We included articles that met all relevant keywords and eligibility criteria aligned with the study’s objective. Our target topic focuses on reviewing the utilization of CAQK in translational models.
2.1 Search strategy
A comprehensive search was performed to identify relevant studies and access the transitional impact of CAQK on different injury models. Articles were obtained from MEDLINE, Embase, Scopus and ScienceDirect databases, covering from inception to July 2024. We employed the search terms “CAQK” and “nanoparticles” as our primary search statement. Other than dates, no limits were utilized in the database search limitations. The initial search identified a total of 93 articles that were published from 2016 to 2024.
2.2 Eligibility criteria
To refine our search, the inclusion criteria focused on papers that implemented CAQK as a therapeutic agent or homing molecule in animal or human studies. We applied the following eligibility criteria on each paper: (1) articles must be written in English; (2) articles must be relevant to the use of CAQK peptide and its therapeutic potential; (3) articles must be randomized controlled trials on treatment outcomes in animals or human subjects; and (4) only medical or biomedical engineering publications are retained. Exclusion criteria were also established to ensure the quality and focus of the systemic review. These criteria include non-peer-reviewed publications, review articles, letters to the editor, meta-analyses, commentaries, conference abstracts, editorials, studies with insufficient clinical data and duplicate publications across different databases. The screening and selection process for articles included in this systemic literature review are illustrated via a PRISMA flow diagram.
2.3 Data extraction
Two reviewers independently reviewed all studies and independently extracted data from the studies using a predefined protocol. Any disagreement between the review authors was resolved through discussion. From the initial pool of 93 articles, a total of 16 articles were deemed eligible for full text review and inclusion in this systemic review. In accordance with recommendations from the Cochrane Handbook for Systematic Reviews of Interventions (chapter 7), data was compiled into a shared Microsoft Excel spreadsheet. The collected data included the name of the paper, year of publication, the list of authors and the publisher, as well as information pertinent to the systematic review including injury model, injury site, route of administration, dose, conjugated nanoparticle, therapeutic agent, cellular impact, functional assessment and side effects. This structured approach ensured a comprehensive review of the available data and information on CAQK, focusing on its translational potential as a therapeutic strategy on injury models. The extracted data was analyzed and discussed with the aim of providing a deeper understanding and insight into the utilization, efficacy, safety, and future applications of CAQK.
2.4 Assessing risk of bias
The quality of studies selected in this systemic review was analyzed and determined by consensus among all authors using The Cochrane Collaboration tool12. Each domain was evaluated for risk bias and rated as “low”, “high”, or “unclear” with provided reasons to support their judgement. The summary of the quality assessment using this tool is summarized in Table 1.
Comment 4: The results section should be improved. Again, you have no data on the assessment and risk of bias in this part.
Author’s response: Thank you and we have added an assessment and risk of bias section to the discussion that speaks to this now. (Lines 459-472)
Change to Text:
4.8 Assessment and Risk of Bias
Any definitive statement is hard to make with a systematic review as it is an interpretation of numerous articles by a reviewer. What one captures for review is limited to what is reported. All these articles published a positive result leading to a publication bias that affects the collective literature on animal experiments using CAQK. They all report the ability of CAQK to home their experimental agent to an injury site. This gives the notion that CAQK is only useful as a homing peptide. As mentioned before, a confounding bias cannot be excluded as CAQK was not ever tested individually and compared against the groups that conjugated CAQK to their nanoparticle containing a therapeutic agent for outcomes. When one experiments, for example, with an experimental agent in saline, a group is normally devised that receives only saline, to rule out any direct impact of saline administration. When this is not done it makes it unclear if any baseline direct impacts, in this case from CAQK, are occurring and unknowingly adjusting outcomes.
Comment 5: The reference list is short. Try to improve with more significant studies, particularly in the discussion section.
Author’s response: Thank you, we have improved on this and now have more references throughout the discussion.
Comment 6: If you do not think this paper can address totally the PRISMA requirement, then you can change the type of article from systematic review to scoping review. Still, my suggestions are relevant and would improve the quality of the present paper.
Author’s response: Thank you and we have incorporated your suggestions to address totally the PRISMA requirements as also requested in your prior comments.
Round 2
Reviewer 1 Report
Comments and Suggestions for Authors
The authors have made essential manuscript changes and responded to my queries. In my opinion, the article is suitable for publication
Author Response
Reviewer 1:
The authors have made essential manuscript changes and responded to my queries. In my opinion, the article is suitable for publication.
Author’s response: Thank you for your thorough review and feedback. We appreciate your time and effort in improving our manuscript. We're glad to hear that the revisions and responses addressed your queries. We look forward to moving forward with the publication process.

Reviewer 2 Report
Comments and Suggestions for Authors
Dear Authors,
Thank you for responding to my requests. There are some other minor points to be improved:
1. Figures 2,3,4 are original? If so, could you offer a more detailed legend? Also, mention the software you have used (ex. BioRender) to create them.
2. Subchapter 3 organ specificity - please be more clear why organ specificity would be important.
Author Response
Reviewer 2:
Comment 1: Figures 2,3,4 are original? If so, could you offer a more detailed legend? Also, mention the software you have used (ex. BioRender) to create them.
Author’s response: Thank you for your comment. Figures 2, 3, and 4 are original and were created using BioRender. We have provided more detailed legends for each figure in the revised manuscript. We appreciate your valuable feedback. (Lines 163-167, 210-213, 216-222)
Change to text:
Figure 2. Injury models and experimental therapy routes of administration. (A) Intravenous injection administered via the tail vein in SCI model. (B) Retro-orbital sinus injection in TBI model. (C) Intra-articular injection into knee joint space in patellar disconnection model. (D) One study utilized local injection at the injury site via a hydrogel application directly to the site of SCI. Created with BioRender.com.
Figure 3. Investigated and postulated CAQK binding sites. A schematic illustration of the extracellular matrix components and their relationship with neuronal surfaces, focusing on potential CAQK binding sites. (HAPLN4 – hyaluronan and proteoglycan link protein 4, CS-GAG – chondroitin sulfate-glycosaminoglycan). Created with BioRender.com.
Figure 4. CAQK conjugated nanoparticles and investigated therapeutic agents. An overview of various CAQK-conjugated nanoparticle systems, designed for delivering different therapeutic agents that have been used in experimental studies. (BFGF – Basic fibroblast growth factor, BDNF – Brain derived neurotrophic factor, DTX – Docetaxel, si/mRNA - small interfering/messenger ribonucleic acid, IFG-1 – Insulin like growth factor 1, TN – Tat-NR2BAc, APNP – Activatable protein nanoparticle, MP – Methylprednisolone). Created with BioRender.com.
Comment 2: Subchapter 3 organ specificity - please be more clear why organ specificity would be important.
Author’s response: Thank you for your continued helpful feedback. In looking at this section more, we agree and have incorporated into the benefits/side effects section. We also restructured the paragraph to focus more on the ability of CAQKs natural excretion to be influenced and how this can impact outcomes/side effect. (Lines
Change to text:
Our review highlighted that CAQK effectively aided in targeting of the desired injury site and conferred therapeutic benefit across nearly all the studies. The benefits ranged from macroscopically measured outcomes, in example, improved motor evoked potentials or some form of motor scoring system, to cellularly measured outcomes, including reduced pro-inflammatory markers to neuroprotection and axonal preservation. Decreased central nervous system lesion size was also noted in 5 studies evaluated. Unfortunately, all studies assessed for these utilized a nanoparticle therapeutic agent conjugated to CAQK, and none directly compared it to CAQK administration only, in regards to benefits/side effects, making for any single benefit/side effect difficult to decipher.
Most benefits or side effects appeared to be unique to the attached respective therapeutic. This is deduced given that no two studies yielded similar results at study conclusion. Regarding side effects, several studies investigated the accumulation/excretion of CAQK conjugated to the nanoparticle therapeutic agent in various other organs. Mann et al. originally showed that no comparable amount of CAQK accumulation was observed in the lung, liver, or skin, after 30 minutes of circulation, except for the kidney, which is recognized as a primary site for peptide clearance. This observation was consistently corroborated by subsequent studies using real time imaging or post-experiment organ harvesting and examination. However, variations in organ accumulation/excretion were observed depending on the nanoparticle therapeutic agent conjugated to CAQK. For instance, 2 studies employing metformin as a therapeutic agent reported higher liver accumulation compared to the kidney, aligning with metformin’s known hepatic action33. In one of those studies, they also go on to describe less liver pathology on histologic examination. These findings are interesting because it shows the ability of using nanoparticles to minimize the toxic effects of a therapeutic agent. This fact is consistent with the literature. Lee et al. demonstrated that a high dose of free methylprednisolone (100 mg/kg) and a low dose of methylprednisolone loaded in nanoparticles (10 mg/kg) have the same effect in reducing astrocyte activation in the spinal cord-injured mice. Therefore, their results showed that the nanoparticle was able to reduce the amount of required drug for diminishing the inflammation37.
It remains unclear if CAQK collection is affected by nanoparticle therapeutic agent attachment, as only these studies directly assessed and compared CAQK to their CAQK conjugated nanoparticle therapeutic agent group. This raises questions on whether the nanoparticle therapeutic agent itself can affect the distribution and excretion of CAQK, and if this is dependent on the concentration or mass ratio of CAQK and the nanoparticle therapeutic agent. This variability could potentially pose a unique predicament, should, for example, CAQK bound to a nanoparticle therapeutic agent become excreted in an organ that is at baseline pathologic, resulting in undesired outcomes that could negate a potential therapeutic benefit. Future research studies should account for these variables when comparing treatment modalities involving CAQK to help results interpretations related to CAQK distribution/excretion and minimize confounding factors.
The duration of studies also varied widely, from hours to weeks before animal euthanasia resulting in numerous limitations such as potential placebo effects, which have been documented in animal studies as well38. Elaboration on the reasoning for each studies respective duration was not given in any study. This variability highlights the need for careful consideration in future experiments design to additionally assess the long term effects and efficacy of CAQK influenced therapies.
